# Robustification of RosettaAntibody and Rosetta SnugDock

**Jeliazko R. Jeliazkov**[1], **Rahel Frick**[2], **Jing Zhou**[2], **Jeffrey J. Gray**[1,2,3,4]*

**1** Program in Molecular Biophysics, Johns Hopkins University, Baltimore, Maryland, United States of America, **2** Chemical and Biomolecular Engineering, Johns Hopkins University, Baltimore, Maryland, United States of America, **3** Sidney Kimmel Comprehensive Cancer Center, Johns Hopkins University, Baltimore, Maryland, United States of America, **4** Institute for Nanobiotechnology, Johns Hopkins University, Baltimore, Maryland, United States of America

* jgray@jhu.edu

**Data Availability Statement:** The models analyzed in this publication and the associated code are both available on Zenodo (doi.org/10.5281/zenodo. 4060853).

**Funding:** This work was supported by the National Institutes of Health (https://www.nih.gov/). JRJ

## Abstract

In recent years, the observed antibody sequence space has grown exponentially due to advances in high-throughput sequencing of immune receptors. The rise in sequences has not been mirrored by a rise in structures, as experimental structure determination techniques have remained low-throughput. Computational modeling, however, has the potential to close the sequence–structure gap. To achieve this goal, computational methods must be robust, fast, easy to use, and accurate. Here we report on the latest advances made in RosettaAntibody and Rosetta SnugDock—methods for antibody structure prediction and antibody–antigen docking. We simplified the user interface, expanded and automated the template database, generalized the kinematics of antibody–antigen docking (which enabled modeling of single-domain antibodies) and incorporated new loop modeling techniques. To evaluate the effects of our updates on modeling accuracy, we developed rigorous tests under a new scientific benchmarking framework within Rosetta. Benchmarking revealed that more structurally similar templates could be identified in the updated database and that SnugDock broadened its applicability without losing accuracy. However, there are further advances to be made, including increasing the accuracy and speed of CDR-H3 loop modeling, before computational approaches can accurately model any antibody.

## Introduction

Antibodies are a crucial component of the adaptive immune system of vertebrates. They are antigen-specific and can be directed towards virtually any antigen to protect us from infections. Their high specificity, in combination with their favorable biophysical properties and pharmacodynamics, have allowed for their development and use as drugs, diagnostics, and research reagents. Antibodies are glycoproteins and are composed of two identical heavy chains and two identical light chains. The isotype is determined by the constant region that dictates effector functions and half life. These constant regions are the same for antibodies of the same isotype. The variable fragments (Fv) on the other hand, are unique to each monoclonal antibody and provide antigen specificity.

was funded by NIGMS grants F31-GM123616 and T32-GM008403. JRJ, RF, JZ, and JJG were funded by NIGMS grant R01-GM078221. The funders had no role in study design, data collection and analysis, decision to publish, or preparation of the manuscript.

**Competing interests:** J.J.G. is an unpaid board member of the Rosetta Commons. Under institutional participation agreements between the University of Washington, acting on behalf of the Rosetta Commons, Johns Hopkins University may be entitled to a portion of revenue received on licensing Rosetta software, which may include methods described in this paper. As a member of the Scientific Advisory Board of Cyrus Biotechnology, J.J.G. is granted stock options. Cyrus Biotechnology distributes the Rosetta software, which may include methods described in this paper. This does not alter our adherence to PLOS ONE policies on sharing data and materials.

Human antibody variable regions consist of a variable light and a variable heavy domain and are extremely diverse, due to V(D)J recombination and somatic hypermutation. These processes result in sequence diversity primarily located in the complementarity determining region (CDR) loops, where the antigen is bound. The CDR 3 loop of the heavy chain (H3) is the most diverse and often particularly important for antigen binding. The remainder of the variable domains is termed framework region and assumes a conserved immunoglobulin (Ig) fold. Antibodies from camelids and cartilaginous fish were found to contain only a variable heavy chain and are referred to as nanobodies, single-domain antibodies, or VHHs.

While the availability of sequence information has increased sharply thanks to high throughput sequencing technologies [1], methods for structure determination have remained low throughput. In order to understand the role of antibodies in disease and to efficiently develop drugs, there is a demand for structural information, both for unbound antibodies and for antibodies in complex with their antigens. Computational prediction of these structures is both attractive and feasible because of the relative conservation of the Ig fold across different antibodies [2]. There are several algorithms for antibody structure prediction, such as ABody-Builder [3], PIGSPro [4], and RosettaAntibody [5]. Across these methods, framework regions are routinely predicted to below 1 Å root-mean-square deviation (RMSD) [6, 7], as they pose a simple homology modeling problem wherein a similar structure can be readily identified by a search within a template database. However, the diverse sequences of the CDR loops result in a variety of conformations, making accurate prediction more difficult. All CDR loops, except the H3 loop, fold into clusters of conformations that are termed canonical conformations [8, 9]. These loops can be predicted within 1 Å RMSD as long as the correct cluster is identified [10, 11]. On the other hand, the CDR-H3 loop does not have a limited set of canonical conformations, necessitating *de novo* modeling and resulting in lower accuracy models.

For certain applications, an antibody model suffices, but often there is interest in further downstream modeling, particularly docking against a target antigen. The antigen adds yet another layer of complexity and even more potential for error, especially as the CDR loops can move to accommodate induced-fit binding [12]. Many software packages are available for protein–protein docking and several of them have modes specific for antibody–antigen docking, these include ClusPro [13, 14], FRODOCK [15], PatchDock [16], HADDOCK [17], and Rosetta SnugDock [18]. The first three methods are global, rigid-body approaches, adopting different docking algorithms. ClusPro and FRODOCK are fast-Fourier transformation (FFT) based. PatchDock decomposes proteins into geometric patches of hotspots and combines geometric hashing and pose clustering to identify interactions. On rigid targets, for which unbound structures are known, these methods tend to perform well. However, using homology models as input or docking flexible targets remains a challenge for these approaches. Methods such as HADDOCK and SnugDock were developed to address the challenge of flexible docking. HADDOCK is an information-driven flexible docking approach that combines a global rigid body search with ambiguous restraints, simulated annealing in torsion space, and minimization in Cartesian space. SnugDock is a local, flexible docking method that refines the CDR loops (including rebuilding the loops) and re-docks the $V_H$–$V_L$ orientation in the context of the antibody–antigen interface. But to yield low-RMSD models, SnugDock requires an input orientation with the paratope close to the epitope, as it is not a global docking approach. Both HADDOCK and SnugDock were recently assessed with respect to other contemporary docking methods. HADDOCK was compared to ClusPro, LightDock, and ZDOCK on sixteen target complexes and generally out-performed the other methods [17]. HADDOCK achieved a 75% success rate (defined as having a model of acceptable quality or better in the top 10, according to the CAPRI quality definition [19]), whereas ClusPro acheived 67.8%. In another recent assessment, with 67 target complexes, ClusPro, ZDOCK, and SnugDock were

compared. Therein, ClusPro achieved a 34% success rate (defined similarly) while SnugDock managed a 77.6% [20], but these rates are not directly comparable as SnugDock was benchmarked on local refinement, not a global search.

Antibody modeling and antibody–antigen docking are fields under active research. Here we report recent developments of RosettaAntibody and SnugDock to improve accuracy of the predicted structures and to make the software more robust and accessible for users and future developers. The template database is now fully automated and can be updated at will, ensuring access to the latest antibody structures in SAbDab [21]. Both RosettaAntibody and SnugDock can now model heavy-chain only antibodies, without any additional flags or specifications. Options for the protocols have been simplified with defaults set based on benchmarks. Constraints have been introduced to improve the quality of models and to allow experimental data to guide modeling. Finally, as these developments were implemented, a set of scientific tests was curated to regularly assess the performance of RosettaAntibody and SnugDock on real-life scenarios.

## Materials and methods

### Template database automation

We developed a Python script to query SAbDab [21], an online antibody database, for the set of sub-3-Å crystal structures. SAbDab pre-processes antibody crystal structures from the PDB and renumbers them according to the Chothia convention [22]. All residue numbers in this manuscript follow the Chothia convention, unless otherwise specified. Based on the information curated by SAbDab, the script then truncates the antibody structures to the relevant structural regions (light chain residues 1–109 and heavy chain residues 1–112). While a crystal structure typically contains a single unique antibody (light chain and heavy chain), there are several structures with multiple distinct antibodies. When multiple chains are present, to avoid ambiguity, we retain the first reported to the SAbDab summary file. If the structure contains only a single light or heavy chain, we retain it. However, if the chain is a single-chain Fv (scFv) (covalently linked light and heavy chain), then it is ignored to limit downstream errors that could arise if the chains are incorrectly assigned. From the truncated structures, sequences are extracted for the regions specified in Table 1 and will be later used in alignments to select structural templates. CDRs containing chainbreaks are omitted during the BLAST database construction. The database is constructed by pooling sequences of the same structural region and length (*e.g.* `database.L1.11` for all sequences of length 11 of the first light-chain CDR) into a single FASTA file, indexed by PDB ID. Each FASTA file is used to build a BLAST database with the `makeblastdb` command. Additionally, the sequences used to construct the database are compiled by structural region and reported to tab-delimited information files for further analysis. Finally, average B-factors for all atoms in each CDR loop and $V_H$–$V_L$ relative orientation metrics are computed, so these values can later be available to quality filters.

The automated database can be generated by running the `create_antibody_db.py` script. A comparison of the last version of the manual template database and the first version of the automatic template database is presented in the results section.

### Enabling nanobody–antigen docking

In the grafting step of RosettaAntibody we removed the requirement for a light chain. Using the flag `-vhh_only` it is now possible to produce heavy-chain only antibody models. Within SnugDock, we now apply a hierarchical kinematic representation (referred to as a `FoldTree`) of the antibody–antigen complex by taking advantage of "virtual" residues. In Rosetta, such residues are ignored during energy calculations, but can be used to describe translations and

**Table 1. Structural region to sequence mapping for RosettaAntibody.**

| Region[a] | Definition[b] |
|---|---|
| CDR L1 | 24–34 |
| CDR L2 | 50–56 |
| CDR L3 | 89–97 |
| FRL[c] | 10–23 |
| | 35–39 |
| | 46–49 |
| | 57–66 |
| | 71–88 |
| | 98–104 |
| CDR H1 | 26–35 |
| CDR H2 | 50–65 |
| CDR H3 | 95–102 |
| FRH[c] | 10–25 |
| | 36–39 |
| | 46–49 |
| | 66–94 |
| | 103–109 |
| Orientation | L5–L104 |
| | H5–H109 |

[a]CDR and framework region definitions in RosettaAntibody. These definitions are used to extract sequences and templates for the homology modeling database. In a modeling task, templates are selected for each region and combined to produce the initial homology model.

[b]The definitions are based on the Chothia numbering convention, but are modified for use in RosettaAntibody.

[c]The FRH and FRL definitions do not exactly complement the CDR definitions as there are additional (non-CDR) loops that are excluded from the frameworks.

rotations. Throughout SnugDock, a single, "universal" FoldTree permitting both $V_H$–$V_L$ and antibody–antigen docking motions is implemented as described in Fig 1.

## Simplified options, new filters, and new constraints

Improvements were made to the options, filters, and constraints within RosettaAntibody and SnugDock. Briefly, we reduced the number of options required to be set by the user in both protocols by setting optimal defaults based on our benchmarking simulations. For the homology modeling stage of RosettaAntibody, we implemented new filters as command-line options to permit the exclusion of specific template PDB files or of cases where the template and query have sequence mismatches involving proline residues in the CDR loops (see Results). Finally, we implemented an automatic glutamine–glutamine (Q–Q) hydrogen bonding constraint in the CDR-H3 loop modeling stage of RosettaAntibody and in SnugDock.

The Q–Q constraint is described by a flat harmonic potential:

$$f(x) = \begin{cases} \left(\dfrac{|x - x_0| - d_m}{\sigma}\right)^2, & \text{if } |x - x_0| > d_m \\ 0, & \text{otherwise.} \end{cases}$$

Here, $x$ is the distance between the donor and acceptor heavy atoms, $x_0$ is the mean observed

## A Simple FoldTree

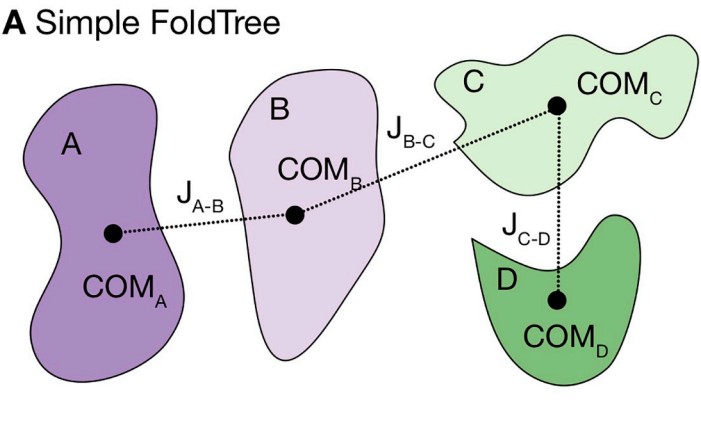

## B Universal FoldTree

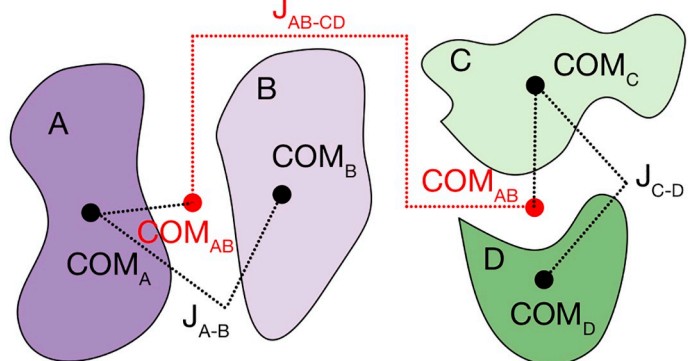

**Fig 1. Comparison of a default `FoldTree` versus one permitting multi-body docking.** Proteins are shown as blobs and labeled A, B, C, and D, jumps (describing relative translations and rotations) are labeled as "J", with subscripts indicating the direction of each jump (*e.g.* A-B indicates that the jump describes the orientation of protein B relative to protein A). Loosely, the ordered collection of all atomic positions and jumps defines a `FoldTree` that can fully describe the motions of a protein complex. In complexes, jumps typically connect the residues closest to their respective protein center of mass (COM). For two chains, perturbing the jump transformation corresponds to simple protein docking. However, when multiple chains are present as in **A**, upstream chains propagate their transformations downstream. In **A**, moving (updating the transformation matrix) $J_{A-B}$ would affect $J_{B-C}$ and $J_{C-D}$, as the jumps are only described relative to one another. The inability to independently move individual proteins or subcomplexes in the default `FoldTree` can limit sampling when large protein complexes are docked. To overcome this, we introduced a hierarchical `FoldTree` (**B**). Here, subcomplexes (*e.g.* antibody chains or antigen chains) have virtual residues placed at their COMs. Jumps connect the virtual residues. The resultant `FoldTree` then contains the transformations relative to the subcomplex COMs and allows subcomplex docking, which was previously not possible.

distance in our antibody database, $d_m$ is the minimal difference at which the penalty will be applied, and $\sigma$ is the observed standard deviation. There are two possible hydrogen bonds between Gln 39 of the heavy chain and Gln 38 of the light chain. We measured the donor–acceptor distances for all antibodies in our updated antibody database that contain the relevant Gln residues. The fit is shown in S1 Fig. The distances between the N and O atoms yielded $x_0 =$ 2.91 Å and $\sigma = 0.23$ Å. The $d_m$ value is chosen to be 0.5 $\sigma$, such that there is no penalty in being within half a standard deviation of the mean and there is a penalty of 0.5 REU at one standard deviation.

### New loop modeling and scientific benchmarks

The final set of improvements to RosettaAntibody and SnugDock brought a new, fragment-based loop modeling approach and new scientific benchmarks to both methods. Briefly, the

new loop modeling method expands on the default kinematic loop closure (KIC) modeling approach in Rosetta [28] by allowing the sampling of backbone dihedral angles from homologous 3- and 9-residue fragments, in addition to the standard sampling from the Ramachandran plot, during loop modeling. After selecting appropriate fragments via the fragment picker [29], fragment KIC can be used during the loop modeling stages of RosettaAntibody and SnugDock. Additionally, we introduced a new form of testing alongside the standard unit and integration tests already present in the Rosetta code base. Scientific testing evaluates the performance of Rosetta on complete scientific tasks (*e.g.* running full antibody modeling simulations to test accuracy rather than just one simulation to test that the code runs without failure). We have made three antibody-related scientific tests: grafting, H3-loop modeling, and antigen docking. The tests are detailed in the subsections below and run automatically on our testing server https://benchmark.graylab.jhu.edu. Both the new loop modeling approach (Pan, X. *et al.*) and the scientific benchmarking framework (Leman, J. K. *et al.*) will be fully detailed in other publications that are currently in preparation.

## Results

### Scientific benchmarking

In the process of developing RosettaAntibody and SnugDock, a series of scientific benchmarks were developed. Scientific benchmarking complements other forms of software testing, such as integration and unit tests, by assessing Rosetta's performance on a diverse set of relevant modeling challenges. A single scientific benchmark consists of a full simulation, whereas unit tests focus on individual functions and integration tests assess exact changes in output. A scientific benchmark is considered successful if the performance is within a certain threshold, usually set by a prior publication. We created three scientific tests for RosettaAntibody and SnugDock. The tests are based on previously published datasets and run regularly on a webserver (https://benchmark.graylab.jhu.edu). There are two RosettaAntibody tests: grafting and loop modeling. While grafting is a fast process ($\leq$ 10 mins per model), CDR-H3 loop modeling is time consuming, so the tests were split based on their runtime.

The grafting test runs the `antibody` executable for 47 targets (listed in the S1 Appendix and available in the antibody database that is distributed with Rosetta), a subset of the 49 targets originally described in [31] (we omit `3m1r` due to its atypically long CDR-L3 loop and `1x9q` as it is an single chain antibody fragment), and it evaluates the RMSDs between the grafted models and the native crystal structures over all antibody structural regions (Table 1).

The CDR-H3 loop modeling test runs the `antibody_H3` executable for a six-target subset of the Marze *et al.* set [23], ranging from easy to difficult, and it evaluates the RMSDs between the models and crystals for the CDR-H3 loop. There is a single SnugDock test that is run on six targets (again ranging from easy to hard) and assesses the interface RSMD between the modeled complexes and the corresponding crystal structures.

### Template database improvements

A homology modeling method, such as the grafting stage of RosettaAntibody, is highly dependent on the structural database it samples for templates. A database with inadequate template coverage will result in poorer modeling outcomes. In the most recent CAPRI assessment [24], we were tasked with modeling two camelid antibodies but could not find suitable non-H3 CDR loop templates in the RosettaAntibody database. Further investigation revealed that the database was outdated and contained artifacts due to its manual curation, a consequence of its initial development in 2008. At that point in time, antibody structures were few and antibody structure databases with consistent numbering schemes, such as IMGT [25], SAbDab [21],

**Table 2. More templates are available for all structural regions in the new database.**

|  | Old Count | New Count | Overlap |
|---|---|---|---|
| **All CDRs** | 1,902 | 2,611 | 1,560 |
| **FRH** | 1,785 | 2,390 | 1,427 |
| **FRL** | 1,577 | 2,832 | 1,111 |
| **Orientation** | 1,003 | 1,721 | 749 |

Comparison of the template count between the last iteration of the manual database and the first iteration of the automatic database (February 15th, 2019). Template counts for each region are shown as well as the "overlap" or number of shared templates between the two databases. Additionally, some sequences in the old database do not appear in the new database because it has more stringent quality criteria. Primarily, 307 structures do not meet the 3 Å resolution cutoff.

and abYsis [26], were not yet developed or, in the case of IMGT, did not use a numbering scheme compatible with RosettaAntibody.

Now we have a new antibody template database that can be automatically generated and updated with the `create_antibody_db.py` script. Table 2 shows the increase in available templates following the update and Fig 2 shows the increase in unique sequence templates for each structural region, which is in the range of 15–49%. We observed that not all the PDBs in the previous template database are retained in the new database, with the specific number varying by structural region. For the entire database, the difference amounts to 342 PDB IDs

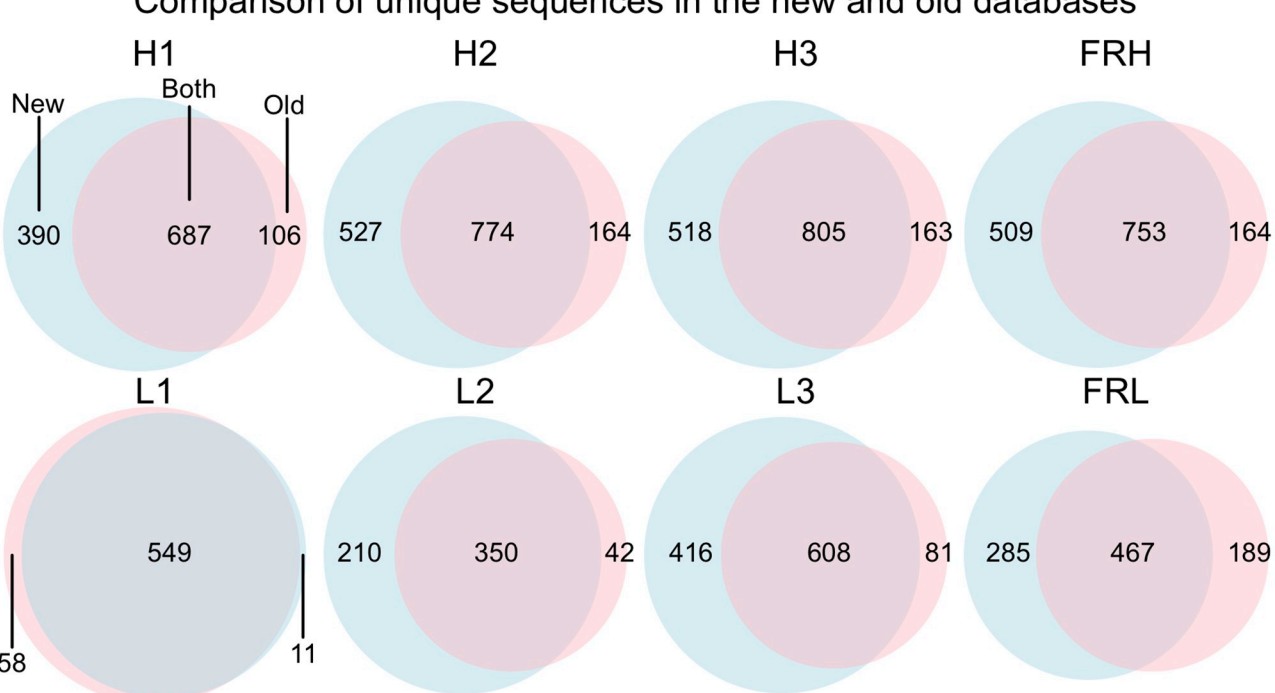

**Fig 2. The new database expands the number of unique sequences for which structural templates are available.** For each structural region, the structures with unique sequences in the old versus new database are compared by Venn diagram. For all regions, there is substantial overlap. The new database always has more sequences than the old database. Some sequences are only found in the old database, this is due to the consistent application of clear selection criteria in the automated database whereas the previous database was manually curated.

that are no longer included. This is due to a few key differences between the two databases. First, we ensure that entire antibody sequences are non-redundant in the new database, no such check was present before. Second, we require that the resolution of structures in the new database is below 3 Å. Again, no such cutoff was present before. Furthermore the previous data included NMR and low-resolution EM structures, which have now been excluded. Of the 342 non-overlapping PDB IDs, the resolution criterion accounts for the most, 307. Third, we have new quality criteria that were not previously implemented, requiring that (1) there are no C–N bond lengths of greater than 2 Å and CA–C–N and C–N–CA bond angles are within the ranges (89.5˚– 144.5˚) and (95˚– 151˚), respectively, (2) all templates can be loaded without error in Rosetta, (3) there are no 0 occupancy atoms in the PDB, and (4) no conserved framework residues, or residues used to identify the different antibody structural regions, are missing. For the overlapping portion of the two databases (identical PDBs), we compared the template structures and sequences to ensure that no drastic changes had occurred. The quality criteria exclude 19 of the 342, which leaves 16 to be excluded by the fact that these PDB records are now obsolete.

Additionally, we found approximately 1–2% of the template regions sourced from the same PDBs mismatched at the sequence level between the old and new database. Investigating the individual cases revealed two general trends. One set of cases arose when multiple antibodies were present in the same PDB asymmetric unit and different antibodies were selected from the multiple possibilities. Another for a sequence mismatch between otherwise identical templates in the new and old database was due to differences in the heavy-chain framework or CDR-H2 loop. In the old database, several variable loops had been incorrectly numbered, possibly because the regular expressions failed to account for edge cases such as engineered antibodies. In the new database, numbering errors are avoided because structures and sequences are derived from Chothia-numbered PDB files, where errors in numbering are minimal [27].

## RosettaAntibody improvements

Improvements to RosettaAntibody affected either the grafting or CDR-H3 loop modeling stage.

**Grafting with an expanded database and filters.**   In the grafting stage, RosettaAntibody benefited from the new template database and new filters. Fig 3 shows a direct comparison of the grafted models for 47 target antibody sequences, previously described in [23]. We omit 3MLR due to its atypical CDR L3 loop. We generated the grafted models as previously described S2 Appendix [31]. We report RMSDs of the loops and framework regions, as well as Orientational Coordinate Distance (OCD), a measure of the relative orientation between the heavy and light chain [23]. In general, we found that the new database produces lower-RMSD grafted models for 53.5–55.0% of target regions. This set of grafting targets was implemented as an automatic scientific benchmark.

Following template selection (based on sequence similarity) potential templates are then filtered based on certain criteria. We introduced a PDB ID and a proline filter to improve the selection process for non-H3 CDR templates. The PDB ID filter excludes a particular PDB from the template set, *e.g.* `-antibody:exclude_pdb 1AHW`. This is useful for benchmarking; if the query sequence has a known structure, then it can be excluded from the template set. The proline filter ensures that prolines match between the query sequences and template structures. Prolines occupy a distinct region of Ramachandran space, but the current template selection approach, BLAST, uses either the BLOSUM62 (for framework alignments) or PAM30 (for CDR alignments) matrix and does not sufficiently penalize proline mismatches. While the filter eliminated proline–non-proline mismatches between template and

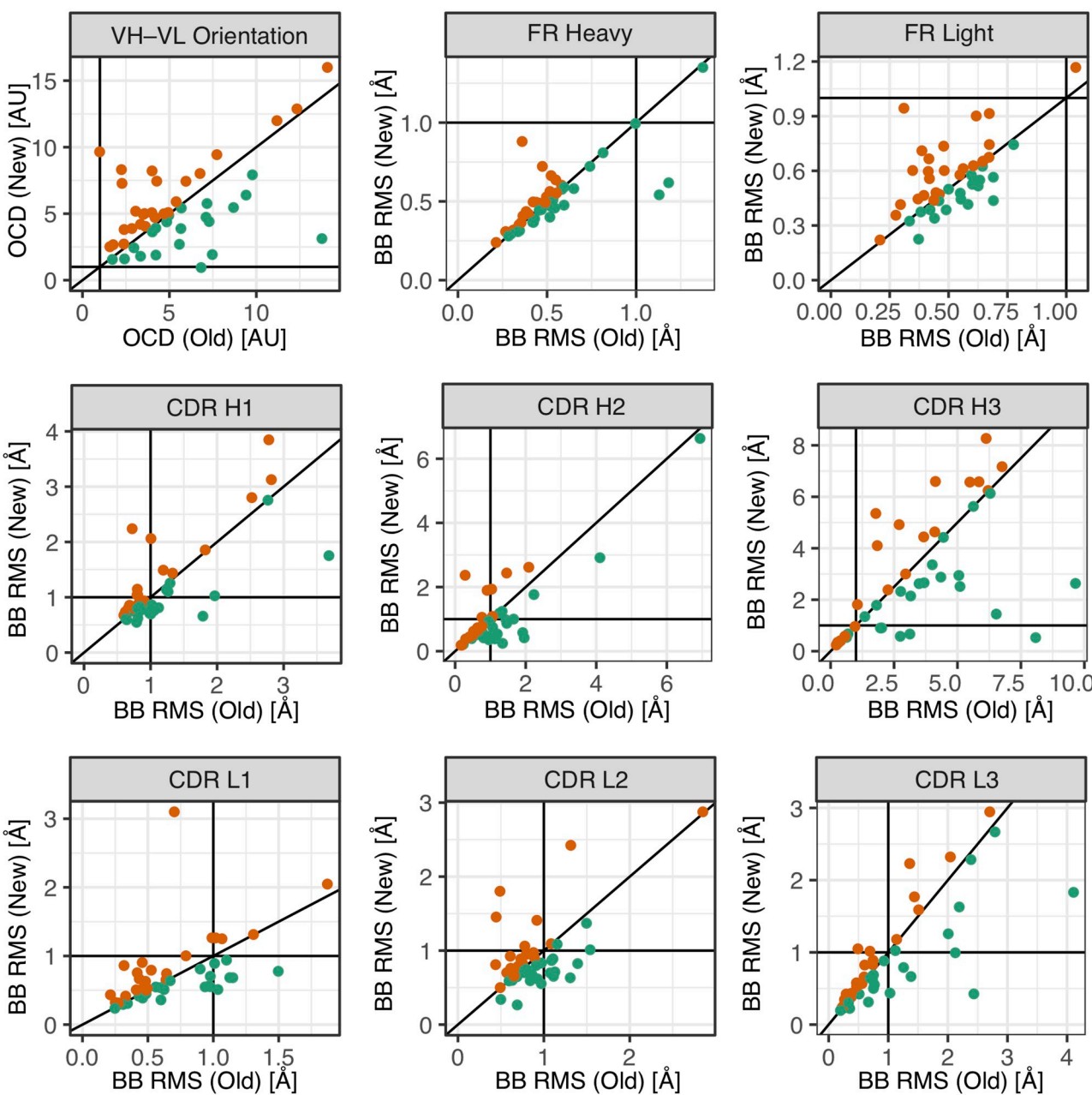

**Fig 3. Comparison of grafted model metrics produced by the new and old databases.** Plots show values for structural metrics (either OCD or RMSD), comparing the grafted models produced using either the old database (manually curated, x-axis) or new database (automatically curated, y-axis) to the crystal structures. Each subplot contains the results for the 47 targets listed in S1 Appendix. Solid lines indicate either 1 Å or 2 OCD and the diagonal (*i.e.* expected performance if there is no change). The OCD is a measure of the heavy-light chain orientation, previously described in [23]. The RMSD is calculated for backbone atoms for the region specified in the subfigure title and defined in Table 1. In the new database, 55% of CDR loops and 53.5% of FRs in the benchmark set have lower RMSD templates than in the old database.

query sequences, it did not demonstrate a concrete improvement in terms of loop RSMD (S2 Fig).

**CDR-H3 loop modeling with fragments and $V_H$–$V_L$ refinement with constraints.** In the CDR-H3 loop modeling stage, we simplified options and introduced a new fragment-based loop modeling method. The options system permits users to pass values to compiled

Rosetta binaries via flags on the command line. To configure the CDR-H3 loop modeling stage of RosettaAntibody, a user previously had to specify the loop modeling method, its settings, and custom constraints to maintain the Q–Q hydrogen bond at the $V_H$–$V_L$ interface, if present. This constraint is now automated and included by default. The legacy options `-cter_insert`, `-flank_residue_min` (bool), `-bad_nter` (bool), `-idealize_h3_stems_before_modeling` (bool), `-remodel` (string), and `-refine` (string) have been completely removed. C-terminal H3 insertions can now be accomplished via fragment-based KIC. We no longer minimize flanking residues during loop modeling or manually adjust CDR-H3 loop dihedral angles, bond angles, and bond lengths, as this does not affect performance. Finally, the `remodel` and `refine` options are removed. These options previously set the loop modeling algorithm, but the loop modeler is now fixed to be KIC, as it has been shown to be the most accurate approach within Rosetta [28]. Furthermore, by refactoring the code to use the newly developed `LoopModel` class, all other loop-related options are by default set to reasonable values, so it is no longer necessary for the user to configure loop-modeling options, although the possibility remains. In sum these efforts have reduced the number of options required to configure RosettaAntibody from approximately 30 to 5 S3 Appendix.

We implemented a new fragment-based loop modeling approach as it was found that fixing sub-regions of loops to match the structures of short fragments (either of length three or nine residues) of similar sequence improved both the fraction of sub-Å models and the RMSDs of near-native models (Pan, X., personal communication). Fragments were selected via the fragment picker on the Robetta server [29]. The new loop modeling method was tested on the 47 antibody targets S1 Appendix and showed no difference in performance when compared to the standard approach. In particular, we expected the use of structural fragments to enhance sampling during loop modeling and lower the minimum RMSD observed across all models. Instead we observed a slight worsening of this metric in the fragment-based models (Fig 4A). As this lack of improvement may have been caused by the highly unique nature of the CDR-H3 loop, we sought to quantify the structural similarity between both protein and CDR-H3 loops and the fragments used in modeling. We investigated the structural similarity between the fragment sets picked for loop modeling and the corresponding target antibody CDR-H3 loop or other (non-antibody) protein loop. For each loop and each possible window of size three or nine residues, the fragment picker selected two hundred selected fragments. These fragments and their corresponding loop segments were compared by measuring the average difference in the backbone dihedral angles as a chord distance (originally defined by Dunbrack and North [8]). We found that non-antibody protein loops were more likely to have near-native fragments identified by the picker than antibody CDR-H3 loops (Fig 4B). This was due to one of two possibilities: (1) either structurally similar fragments exist and the fragment picker cannot identify them for antibody CDR-H3 loops or (2) the fragments do not exist. Considering that the fragment picker tends to perform well across a diverse set of targets [29] and previous observations that antibody CDR-H3 loops have fewer structurally similar fragments in the PDB than other protein loops [30], we concluded that the latter is most likely and the lack of structural similarity between fragments and CDR-H3 loops can explain the inability of fragment-based loop modeling to improve CDR-H3 loop models.

To ensure continuous testing of the CDR-H3 loop modeling stage, we implemented a subset of the Marze *et al.* antibody targets as a scientific benchmark. Specifically, we selected six targets of varying difficulty, based on prior modeling performance [31] and CDR-H3 loop length (S1 Table). The scientific benchmark then consists of running the CDR-H3 loop modeling stage on homology models of these antibody frameworks (S3 Appendix).

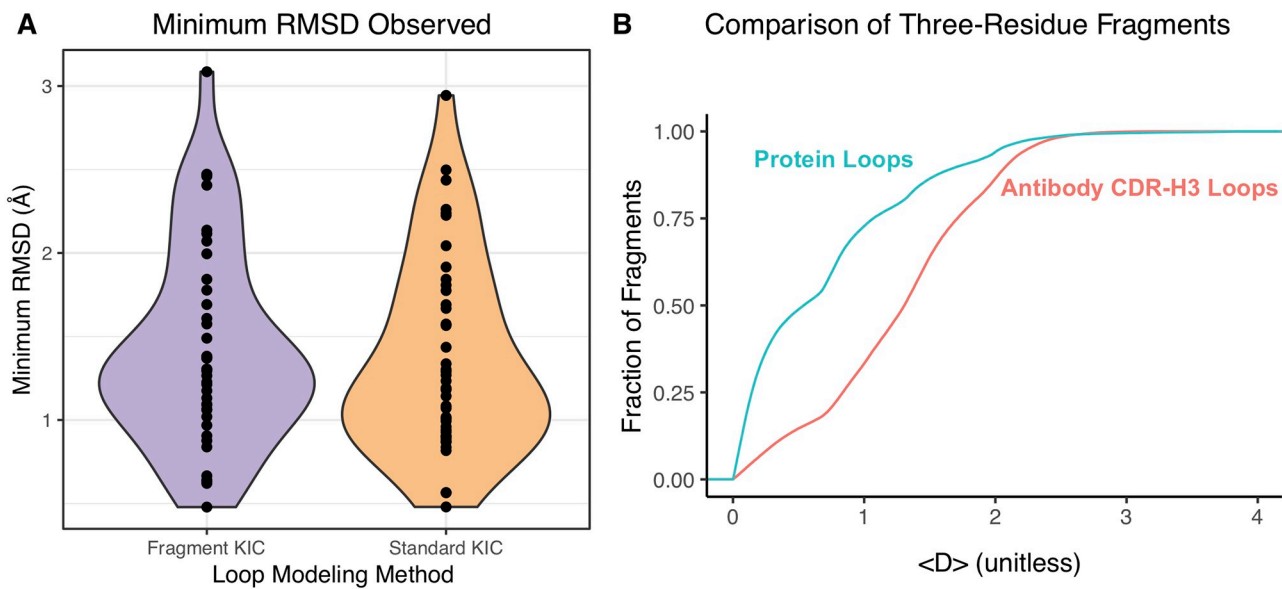

**Fig 4. Comparison of loop modeling methods.** (A) The distributions of the minimum CDR-H3 loop RMSDs observed for all antibodies in the benchmark, for two loop modeling methods, do not significantly differ according to Student's t-test (p-value = 0.67). (B) Three-residue fragments from the PDB are more structurally similar to protein loops than to antibody CDR-H3 loops. All three-residue fragments selected by the fragment picker were compared to their corresponding loop sub-regions. For each fragment and loop combination, a chord distance was calculated to compare the difference in dihedral angles: $\langle D \rangle = \frac{1}{n}\sum_n (D_\phi^2 + D_\psi^2)/2$ where $D^2(\theta_1, \theta_2) = 2 - 2\cos(\theta_2 - \theta_1)$. Thus, $\langle D \rangle$ has a minimum of 0, if a fragment matches a loop exactly, and a maximum of 4, if a fragment differs by 180 degrees at every dihedral angle. The cumulative distribution function of these distances then yields the probability (y-axis) that a fragment is within a certain chord distance (x-axis) of a loop.

Finally, beyond enabling a new loop modeling approach, we introduced an automated $V_H$–$V_L$ Q–Q hydrogen bond constraint. Constraints modify the Rosetta score function by adding customizable functions to the standard collection of physical and statistical terms. A typical use case for constraints is to incorporate experimental data in simulations by penalizing protein conformations that are nonconcordant. RosettaAntibody recommends constraining the C-terminal CDR-H3 loop kink and a Q–Q hydrogen bond at the $V_H$–$V_L$ interface, if present. The kink and the Q–Q hydrogen bond are both present in 81.1% and 88.5% of antibodies in our database. Thus both constraints should be enabled by default. However, the kink constraint was only recently automated [31] and the Q–Q constraint remained user specified until this publication. As a consequence, the constraints were under utilized because they relied on manual user input to identify the corresponding residues and determine the functional form and weights of the constraint.

We implemented a constraint automation similar to the one used by Weitzner and Gray to constrain the kink [31]. Key residues are automatically identified by relying on known sequence features and implementing a consistent numbering scheme throughout modeling. The functional form and weights of the constraint are based on observed geometries in the protein data bank. Using the recently established scientific benchmarking framework, we tested multiple constraint functions and strengths to identify a reasonable default. We found that the harmonic constraint improved the fraction of models in which the hydrogen bond is formed (S3 Fig), but did not significantly affect the CDR-H3 loop RMSDs (S4 Fig). The constraint is now enabled whenever the requisite glutamine residues are present in the antibody sequence.

## Rosetta SnugDock improvements

The primary improvement to SnugDock was the introduction of a more general `FoldTree` that enabled the modeling of heavy-chain only antibodies. Additionally, we introduced the possibility for fragment-based loop modeling, the capacity for experimental constraints, as well as two automated constraints (as in RosettaAntibody), and scientific benchmarks. While outside the scope of this publication, some readers may find a full benchmark and comparison to other software useful. A recent study comparing SnugDock, ZDOCK, and ClusPro is available [20]. Readers may also find useful another recent study comparing ClusPro, LightDock, ZDOCK, and HADDOCK [17].

**FoldTree simplification.**   Primarily, we improved the kinematics of Rosetta SnugDock. The kinematic layer of Rosetta controls how atomic coordinates are updated over the course of a simulation. It is necessary because Rosetta uses internal coordinates (dihedral angles, with fixed bond lengths and angles) to accelerate sampling in most protocols (simulations in Cartesian coordinates are possible, but not common) [32]. Central to the process of keeping internal and Cartesian coordinates up-to-date is an object known as the `FoldTree`, at the residue level, and the `AtomTree`, at the atomic level [33]. The `FoldTree` is implemented as a directed acyclic graph that propagates coordinate changes. For example, a typical `FoldTree` for a four-protein complex would be linearly ordered, taking the chain order from the PDB file (Fig 1A). In this `FoldTree`, one cannot dock a middle protein independently of its neighbors. This poses a problem in the case of an antibody–antigen complex, where the relative $V_H$–$V_L$ orientation might change as the antibody accommodates the antigen. This problem is further amplified when modeling loops, as loops require alterations to the `FoldTree` to permit the repeated breaking and closing of covalent bonds. The typical solution is to switch between multiple, incompatible, "simple" `FoldTree` objects that rely on assumptions about the input and have to be specified beforehand. To overcome this issue, we generalized the set of assumptions applied in the `FoldTree` construction stage of SnugDock, resulting in a single, consistent `FoldTree` that can be used throughout the simulation. This tree also enabled the modeling of heavy-chain only antibodies (*e.g.* camelid).

In the initial implementation of Rosetta SnugDock, it was assumed that the docking partners consisted of a light chain, a heavy chain, and an antigen, in that order. The `FoldTree` was updated at each stage of the simulation to accommodate appropriate sampling. The light chain could be docked to the heavy chain to refine the orientation. In the stage sampling the Ab–Ag interface, the `FoldTree` was re-ordered to have the antigen first then the light and heavy chains, so the antigen could be docked to the antibody. Additionally, during H3 and H2 loop modeling stages, a third `FoldTree` was applied to permit opening and closing the loops. This scheme assumed the presence of a light chain, excluding heavy-chain only antibodies from SnugDock simulations.

To correct this issue, we introduced a more hierarchical `FoldTree` that exploits "virtual" residues—residues that are chemically and physically ignored, but tracked by the `FoldTree` to store positional information. The virtual residues are placed at individual protein and complex centers-of-mass and then connected to corresponding polypeptide chains in a hierarchical fashion (Fig 1B), such that complexes of interest are grouped together (*e.g.* the two antibody chains or any number of antigen chains). Using virtual residues overcomes the aforementioned challenges. First, by placing the proteins downstream of virtual residues, each chain can have its own internal `FoldTree` without affecting any downstream partner. This permits `FoldTree`-dependent modifications within in each chain (such as loop modeling) to take place, without necessitating a new `FoldTree`. Second, by placing virtual residues at the centers-of-mass of each protein and the relevant complexes, simultaneous docking between

multiple partners is now possible in one `FoldTree`. Finally, this `FoldTree` makes no assumptions about the identity of individual chains, so it is compatible with heavy-chain only antibodies.

The new `FoldTree` enabled our participation for Targets 123, 124, and 160 in the blind protein docking challenge called CAPRI, detailed in [24]. Briefly, we ran standard ensemble SnugDock simulations (S5 Appendix). The results showed that we were technically able to model the camelid antibodies, but the models were inaccurate due to the challenges associated with modeling longer CDR-H3 loops (11–21 residues).

**Introducing constraints to SnugDock.** We also implemented automatic Q–Q and kink constraints in SnugDock, and further enabled user-defined constraints. Experimental or computationally-derived epitope data (e.g. [34]) can now guide docking. As a proof of principle, we combined hydrogen exchange-mass spectrometry (HX-MS) data with SnugDock. HX-MS measures the backbone amide hydrogen/deuterium exchange rate, and interacting residues, such as those at epitope or paratope, will yield slower exchange rates that can then suggest binding sites for docking. During the docking process, constraints based on pre-processed HX-MS data are applied to the antibody–antigen complex. Interactions that satisfy the experimental constraints are rewarded, whereas the interaction that violate the constraints are penalized. We derived a constraint form for each antigen-residue suggested by HX-MS to the closest antibody CDR residue by using the so-called `KofNConstraint` with a flat harmonic potential. A `KofNConstraint` adds the $K$ lowest values of a total of $N$ constraints to the score, where the $N$ constraints are for each residue in the paratope.

As a proof-of-principle, we selected a camelid antibody–ricin complex, 5BOZ [35], to evaluate the utility of constraints. This PDB structure is one of several antibody–ricin complexes for which HX-MS data is available (Weiss, D. D., personal communication). We introduced the data in SnugDock as `KofNConstraints` (S9 Appendix). We then ran a local ensemble docking simulation in SnugDock (S10 Appendix) and global rigid-body docking simulation with RosettaDock (S11 Appendix) [36], both constrained based on the HX-MS data. We found that, when starting from the bound crystal structure, the global search with constraints produced low-scoring (favorable) models of high quality (according to CAPRI criteria), (Fig 5). When using SnugDock and starting from a modeled antibody and unbound antigen crystal structure, constraints did not result in high quality models. Interestingly, these models were able to produce native-like CDR-H3 loop structures. A full study on the utility of constraining antibody–antigen docking simulations with HX-MS constraints is currently in preparation (Zhou, J., Weis, D. D. & Gray, J. J.).

## Discussion

Here we presented several advancements in RosettaAntibody and SnugDock that improve performance and collectively lay a foundation for further work. We improved the homology modeling stage of RosettaAntibody by (1) automating the template database to increase coverage and reduce errors, and (2) introducing new filters. We advanced the CDR-H3 loop modeling stage by introducing a new loop modeling approach and structural constraints. We updated SnugDock to use a universal `FoldTree` that enabled the docking of single-domain antibodies, added a new loop modeling protocol, and introduced new constraints. Finally, we implemented scientific benchmarks that regularly test the performance of these protocols.

However, major challenges remain that could be the focus of future development: CDR-H3 modeling, a truly universal `FoldTree` for multi-body docking, and improved selection of non-H3 CDR loop templates. Of these, CDR-H3 loop modeling is the most challenging. Broadly, modeling challenges are binned into two categories: scoring and sampling. We

**Rigid-body Docking Interface RMSD**

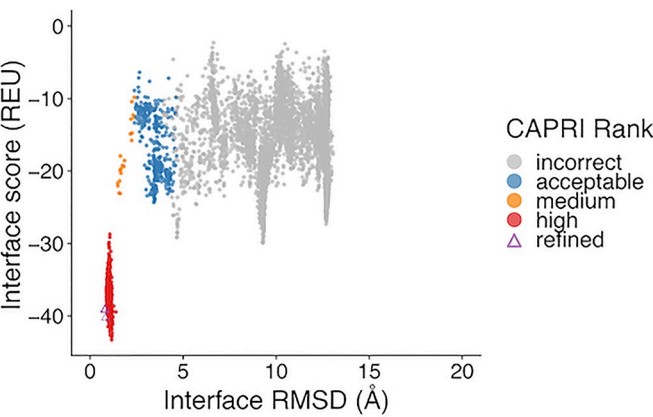

**SnugDock Interface RMSD**

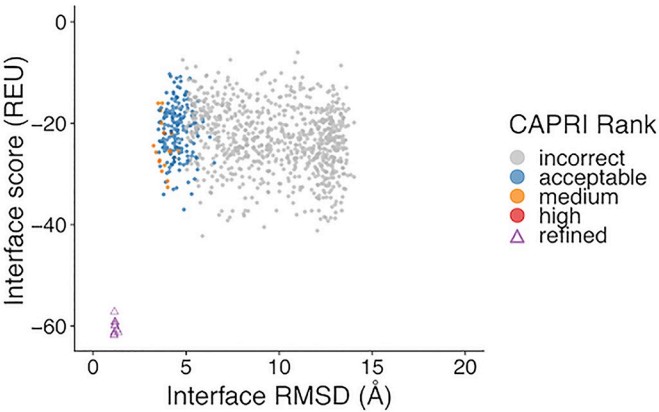

**SnugDock CDR-H3 Loop RMSD**

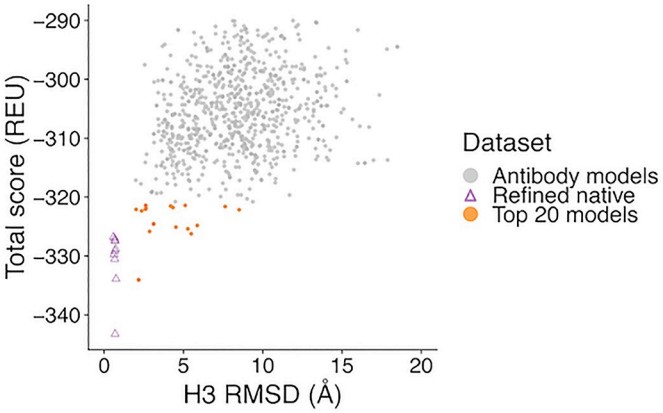

**Fig 5. Constraints aid global antibody–antigen docking, but do not affect local refinement.** A global, rigid-body search with constraints added to the score function resulted in multiple high-quality models, according to the CAPRI criteria (top panel). However, applying the same constraints to a local search with an antibody homology model did not improve sampling (middle panel). Interestingly, the addition of constraints to SnugDock led to the sampling of native-like CDR-H3 loops, despite not including any constraints during the H3 modeling stage of Rosetta Antibody (bottom panel). "Refined" indicates models created from the bound complex structure, for reference.

recently showed that native-like antibody loops, when sampled, can be identified by score alone in Rosetta [31]. We also observed that for some targets it is challenging to observe a native-like conformation in the set of all models [6, 24]. Thus, the CDR-H3 loop modeling problem is primarily a sampling challenge. The anecdotal evidence is further supported by observations that CDR-H3 loops are exceptionally diverse, as has been previously demonstrated by others [30] and shown by us here (Fig 4B). One possible approach to overcoming the sampling challenging is to accelerate the loop modeling step to sample more loop conformations. As the slow stage of generating loop models is scoring and filtering, using a knowledge-based rather than physical potential may provide a viable alternative. For example, KORP is a potential capable of scoring 100,000 12-residue loop decoys in under a minute [37]. Another approach to improving sampling would be a more specialized fragment insertion routine during loop modeling. The method used here relied only on sequence similarity to select fragments of either length 3 or 9 and inserted the fragments randomly throughout the loop modeling simulation. An alternative fragment selection approach would not restrict fragment size and might choose fragments from CDR-H3 or H3-like loops. Fragment insertion would focus on the termini that are more structurally conserved regions, *e.g.* approximately 90% of antibodies have a C-terminal "kink". Finally, emerging deep-learning-based approaches may accelerate CDR-H3 loop sampling [38]. The new loop modeling framework has laid the foundation for exploring further strategies.

The hierarchical `FoldTree` introduced here allows more flexibility in SnugDock and enable the docking of single-domain antibodies. However, true multi-body docking is still not possible as the SnugDock approach is a specialized class, separate from the general docking approach in Rosetta. Moving forward, docking approaches in Rosetta should be unified. *I.e.* the `DockingProtocol` class should be able to provide all docking functionality, based on user specifications and input.

Finally, the homology modeling stage of RosettaAntibody relies on BLAST to select structural templates for query sequences for the various structural regions of an antibody (Table 1). However, most structural regions are small while BLAST is not optimized for aligning short sequences. Thus going forward we must consider alternative approaches to alignment such as custom PSSMs or machine-learning-based approaches [10, 11].

## Conclusion

The role of computational modeling will grow as the throughput of experimental techniques continues to increase. To enable the continued development of the RosettaAntibody and SnugDock protocols, we have simplified their usage, robustified their performance on varied targets, and developed scientific benchmarks. By simplifying the usage of these protocols, future developers can focus on improving the underlying algorithms rather than fiddling with extraneous options. Increasing the utility of these protocols will ensure their longevity as increasingly diverse and challenging pathogens lead to the development and discovery of atypical antibodies. Finally, the availability and regular assessment of scientific benchmarks will encourage a more rapid developmental cycle.

## Supporting information

**S1 Fig. Q–Q hydrogen bond distances observed in the RosettaAntibody database. Left:** The histogram depicts the observed distances between the oxygen and nitrogen atoms of light chain residue Q38 and heavy chain residue Q39. The distribution was fit by kernel density estimate using Gaussian kernels. **Right:** The negative logarithm of the probability is proportional

to the energy. A harmonic function was fit in the range of 2.5 Å to 3.1 Å.
(TIFF)

**S2 Fig. Proline filter has minimal effect on grafted model RMSDs.** Comparison of the non-H3 CDR loop RMSDs before and after the application of a proline filter. The filter prevents the use of a template when there is a mismatched proline residue with the query. The differences show that most loops are unaffected. In one case for the CDR H2 loop, the loop is model is worse following the application of the filer (moving 2 Å further from the native). This is exclusively due to the presence of an glycine at the start of the target loop (PDB ID: 3LMJ). In the initial model (PDB ID 6EIK, no proline filter), the template also has a glycine, correctly modeling the initial loop structure, whereas the proline-filter-selected template (PDB ID 5LSP) lacks this initial glycine and cannot accurately model the loop start resulting in a cascading worsening of the loop model. All other loops show minor variations within 1 Å.
(TIFF)

**S3 Fig. The Q–Q constraint increases the fraction of models that form hydrogen bonds.** We generated 500 decoys of 6 antibodies with solved structures (S1 Table) either without or with a flat harmonic constraint between the relevant Gln residues. Left: The distances between the nitrogen and oxygen atoms of residues Q38 of the light chain and Q39 of the heavy chain were measured and compared to the native distributions in our antibody database. Right: Each decoy was analyzed for presence of the two possible hydrogen bonds using PyRosetta's get_h-bonds() function. The fraction of decoys forming both hydrogen bonds is shown for each antibody (color-coded).
(TIFF)

**S4 Fig. The Q–Q constraint does not appear to have a strong effect on CDR-H3 loop modeling.** A funnel plot (total score versus CDR-H3 loop RMSD) comparison of RosettaAntibody on six benchmark antibodies does not show a significant difference after the incorporation of the Q–Q constraint. The constraint seemingly improves performance on targets 2VXV and 4F57, but worsens it on 3M8O.
(TIFF)

**S1 Table. Target antibody CDR-H3 loops for the antibody modeling scientific benchmark.**
(PDF)

**S2 Table. Target antibody–antigen complexes for the docking scientific benchmark.**
(PDF)

**S1 Appendix. List of antibodies used in the grafting benchmark.**
(PDF)

**S2 Appendix. RosettaAntibody grafting command line for benchmarking.** The command line below is used only to compare the grafting assembly stage of RosettaAntibody. In a genuine run, the `-no_relax` would not be used and `-antibody:n_multi_templates 10` would be used instead.
(PDF)

**S3 Appendix. RosettaAntibody CDR-H3 loop modeling command line.** Note constraints are now automatically enabled. To disable constraints, use `-antibody:constrain_vlvh_qq false`, `-antibody:h3_loop_csts_lr false` and `-antibody:h3_loop_csts_hr false`.
(PDF)

**S4 Appendix. RosettaAntibody CDR-H3 loop modeling command line with fragments.**
Note constraints are now automatically enabled. To disable constraints, use `-antibody:`
`constrain_vlvh_qq false`, `-antibody:h3_loop_csts_lr false` and
`-antibody:h3_loop_csts_hr false`.
(PDF)

**S5 Appendix. SnugDock command line.** Note constraints are now automatically enabled. To
disable constraints, use `-antibody:constrain_vlvh_qq false`, `-antibody:`
`h3_loop_csts_lr false` and `-antibody:h3_loop_csts_hr false`.
(PDF)

**S6 Appendix. SnugDock command line with an ensemble of structures.** Note constraints
are now automatically enabled, to disable constraints, use `-antibody:constrain_`
`vlvh_qq false`, `-antibody:h3_loop_csts_lr false` and `-antibody:`
`h3_loop_csts_hr false`. Furthermore, structures must be prepared for ensemble dock-
ing by `docking_prepack_protocol` see (below).
(PDF)

**S7 Appendix. Prepack protocol command line.** This will alter the `antigen.list` and
`antibody.list` files in place. Please note that the chain order in the `-partners` flag
must match the order of chains in the PDB passed by the `-s` flag and `-ensemble1` and
`-ensemble2`. That is to say in the example below the `initial_conformation.pdb`
file has the "A" chain first followed by "H" and "L" while the first ensemble is a list of antigen
only structures and the second ensemble is a list of antibody only structures. All structures
must have matching numbers of residues.
(PDF)

**S8 Appendix. Sample list file.** The ensemble of antibody sturctures in this case comes from
differ H3 models, but ensembles can also be generated by FastRelax, for example.
(PDF)

**S9 Appendix. Sample KofNConstraint file.** This file only contains two constraints as an
example. A complete file would contain one `KofNConstraint` for each antigen residue
with HX-MS data. Each `KofNConstraint` would contain one flat harmonic constraint for
each CDR residue.
(PDF)

**S10 Appendix. SnugDock command line with constraints and motif dock score (MDS).**
Additional constraints can be added to both the low- and high-resolution stages of SnugDock.
MDS is a special score function for the low-resolution stage of docking. It has been found to
improve performance in protein–protein complex docking. It can be used in SnugDock as
well.
(PDF)

**S11 Appendix. Global docking command line.** Exemplary flags for global docking with con-
straints.
(PDF)

## Acknowledgments

The authors of this study would like to acknowledge Drs. Nicholas Marze and Brian Weitzner
(both of Johns Hopkins University) for helpful discussions and advice, Xingjie Pan and Prof.
Tanja Kortemme (both of University of California San Francisco) for help with implementing

the refactored loop modeling code for antibody modeling, and Dr. Julia Koehler Leman (Flatiron Institute) and Sergey Lyskov (Johns Hopkins University) for implementing the benchmarking server. Prof. David Weis (University of Kansas) advised the work including HX-MS data.

## Author Contributions

**Conceptualization:** Jeliazko R. Jeliazkov, Jeffrey J. Gray.

**Funding acquisition:** Jeffrey J. Gray.

**Investigation:** Jeliazko R. Jeliazkov, Rahel Frick, Jing Zhou, Jeffrey J. Gray.

**Methodology:** Jeliazko R. Jeliazkov.

**Resources:** Jeffrey J. Gray.

**Software:** Jeliazko R. Jeliazkov, Rahel Frick, Jing Zhou.

**Supervision:** Jeffrey J. Gray.

**Validation:** Rahel Frick, Jing Zhou.

**Visualization:** Rahel Frick.

**Writing – original draft:** Jeliazko R. Jeliazkov, Jeffrey J. Gray.

**Writing – review & editing:** Jeliazko R. Jeliazkov, Rahel Frick, Jing Zhou, Jeffrey J. Gray.

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
