## [Decision Letter · Decision Letter 0]

21 Sep 2020

PONE-D-20-14845

Robustification of RosettaAntibody and Rosetta SnugDock

PLOS ONE

Dear Dr. Gray,

Thank you for submitting your manuscript to PLOS ONE. After careful consideration, we feel that it has merit but does not fully meet PLOS ONE’s publication criteria as it currently stands. Therefore, we invite you to submit a revised version of the manuscript that addresses the points raised during the review process.

Thank you for your patience during the review process. Your study has been reviewed by four experts in the field. Please provide a response letter, answering point-by-point all the comments raised by the reviewers.

We look forward to receiving your revised manuscript.

Kind regards,

Horacio Bach

Academic Editor

PLOS ONE

Additional Editor Comments:

Dear Prof Gray

I have now the revisions for your manuscript. As you can see both reviewers are very supportive of your work. I classed however as Major revision because I think the comment by Reviewer 1 on comparing your approach against an existing methodology on a small dataset (16 complexes) has merit. Given the small size of the set I believe this could be done very quick and it would provide an additional piece of data to better position your method among existing methodologies. There is also a number of minor issues that I invite you to consider too.

Journal Requirements:

2. Thank you for including your competing interests statement; "JJG is an unpaid board member of the Rosetta Commons. Under institutional participation agreements between the University of Washington, acting on behalf of the Rosetta Commons, Johns Hopkins University may be entitled to a portion of revenue received on licensing Rosetta software, which may include methods described in this paper. As a member of the Scientific Advisory Board of Cyrus Biotechnology, JJG is granted stock options. Cyrus Biotechnology distributes the Rosetta software, which may include methods described in this paper."

Reviewers' comments:

Reviewer's Responses to Questions

**Comments to the Author**

1. Is the manuscript technically sound, and do the data support the conclusions?

Reviewer #1: Yes

Reviewer #2: Yes

2. Has the statistical analysis been performed appropriately and rigorously? 

Reviewer #1: No

Reviewer #2: Yes

3. Have the authors made all data underlying the findings in their manuscript fully available?

Reviewer #1: No

Reviewer #2: Yes

4. Is the manuscript presented in an intelligible fashion and written in standard English?

Reviewer #1: Yes

Reviewer #2: Yes

5. Review Comments to the Author

Reviewer #1: The paper is solid and well-written. The Gray group is well known and pioneers in the field of macromolecular docking of highly flexible partners. In addition, improvements in the Rosetta code is always welcome by the scientific community, and in particular for antibody design which is now of the highest relevance due to the coronavirus pandemic.

I have only one major comment that must be considered before publication and some minor comments to improve readability and reproducibility of the great work reported in this manuscript.

Major comment

Overall, the authors should compare docking results of antibody-antigen complexes to those retrieved by HADDOCK (https://doi.org/10.1016/j.str.2019.10.011) using the same benchmark of 16 complexes from the docking benchmark. It would be extremely useful to discuss those as well in the light of Rosetta/SnugDock performance because the aforementioned paper includes comparison of 4 different software. In addition, the authors should improve the illustration of data in all main figures and share their models, calculations and statistical analysis via the SBGrid database or another resource, in order for the work to be reproduced by others.

Minor comments

1-Text

1.1-Page 5, line 132: section “New loop modeling and scientific benchmarks”. I understand the authors would like to publish those results elsewhere, but at least they should explain briefly what they did.

1.2-Page 6, line 156: The authors state that their previously published benchmark contains 48 targets, but in the previous publication 46 targets were included. I would recommend the authors to share along with Appendix S1 all pdb input files that were used for calculations as a zip file or deposit in SBGrid as recommended above.

2-Figures.

2.1-Figure 1: Figure is good, but the old is shown of being of similar importance to the new FoldTree. Please show that the new one is the relevant one so that the reader knows directly by looking at the figure, that this is the one that was developed in this manuscript.

2.2-Figure 2: Please include more details in the figure, explain what are the unique sequences for each plot in the legend; show the overlap of sequences in the old vs new databases with a venn diagram for each or another representation; show with additional figure panels example of interesting new data per category.

2.3-Figure 3. Can you please make the figure legends more explanatory? Currently, they are called “Old Value” and “New value” instead of OCD (Å) or RMSD (Å). Results to my opinion, do not seem significant and authors cannot state that there is an improvement in grafted models as shown by these metrics. Authors should color-code the plots, showing with e.g. blue the grafted that were improved, black those with no change, and red those that worsened. In addition, statistics are missing, e.g. how many data points are per plot? Which exact models are included? The authors should share (or deposit in SBGrid as recommended above): (a) the old, the new grafted models and reference models plotted via the supplementary material as a zip file, (b) the script to calculate OCD so that plot results can be reproduced and (c) describe what is this exact RMSD that is calculated (main chain atoms? All chain? which exact regions?)

2.4-Figure 4. Please change the colors of the figure because at the beginning I thought that plot B colors correspond to violin plot colors, which is not the case.

3-Tables

3.1-Table 2. Is there a reason why on Table 2 there are ~300-400 templates removed from the old database? The authors should explain this in-text. What are these stringent quality criteria the authors mention in the Table legend? Only missing atoms or non-realistic C–N distances in critical regions? Please explain in-text or in a Table the old and the new criteria.

Considering the figure/table optimization and comparison to the other docking software, I believe this paper will be a seminal reference for future work regarding antibody design.

Reviewer #2: Reviewer Comments

The manuscript titled “ Robustification of RosettaAntibody and Rosetta SnugDock” describes developments made to the existing RosettaAntibody and Rosetta SnugDock softwares for antibody modelling and antigen-antibody docking, respectively. The advances made to each of these softwares are briefly summarized below.

For RosettaAntibody, the improvements include i. the automation of generating the template database by extracting PDB structures from another existing database- the Structural Antibody Database (SAbDab). The antibody structures extracted from the SAbDab, used for antibody modelling purposes, are numbered according to the Chothia scheme. ii. elimination of the requirement for a light chain during the grafting step of RosettaAntibody. ii. Minimizing user intervention by reducing the number of user-specified options during the homology modelling stage of RosettaAntibody and setting of default options (using the fragment-based kinematic loop closure (KIC) by default, to model C-term CDR-H3 insertions). iii. Options to exclude certain template PDB files or templates where the template and query sequences differ by the presence of proline residues in the CDR loops. iv. Incorporation of a glutamine-glutamine hydrogen bonding constraint during the modelling of the CDR H3 loop.

In case of SnugDock, the following changes made are i. Introduction of virtual residues and a modification to the existing kinematic representation (referred to as the “FoldTree”) to improve the flexibility of VH-VL and antigen-antibody docking motions. ii. automatic inclusion of the glutamine-glutamine hydrogen bonding constraint and kink constraint during the CDR-H3 modelling stage. iii. Docking of antigen to nanobodies. iv. Fragment-based loop modelling, as done for RosettaAntibody. v. Enabling usage of hydrogen exchange-mass spectrometry (HX-MS) data in the form of constraints to improve accuracy of the docking (detailed in another publication)

Apart from these modifications, the authors have also developed new scientific benchmarks for testing these methods, which is being described in another publication.

The authors tested the performance of the updated softwares using datasets published previously and compared the results with the same tests done on the earlier version of the softwares (including the earlier template database). The grafting step and loop modeling step of RosettaAntibody, and the docking step of SnugDock were tested.

The paper is well-written and balanced. Improvements as well as the limitations of the updated softwares have been detailed clearly. They have also described cases in which the updates did not have any significant improvement on the tests.

I have a few minor suggestions.

1. Does changing the default loop modelling step during CDR-H3 modelling to the kinematic loop closure method increase or decrease the time taken at the CDR-H3 modelling stage? The CDR-H3 modelling step has been tested on six antibody targets with a range of difficulties. Is it possible to run this step on slightly more number of antibody targets and make a comparison of the time (in addition to comparing the accuracy which has already been done) taken with the previous loop modelling and the kinematic loop closure method?This will give readers an idea of how long it would take to complete this specific step, as this is the limiting step during the modelling process.

2. Do the softwares always utilize all the processors it has been allocated, or does it sometimes use only a single processor despite multiple processors being allocated to it, if using a non-MPI version? What kind of system requirements will enable the non-MPI version of the softwares to make use of all the processors granted to it? A line or two about this could be included in the manuscript.

3. In Figure 5, the colors in the legend are not clear (at least in the PDF version), so maybe try to make the sizes a little larger.

6. PLOS authors have the option to publish the peer review history of their article (what does this mean?). If published, this will include your full peer review and any attached files.

Reviewer #1: No

Reviewer #2: **Yes: **Jarjapu Mahita

---

## [Author Response · Author response to Decision Letter 0]

18 Dec 2020

Reviewer #1: The paper is solid and well-written. The Gray group is well known and pioneers in the field of macromolecular docking of highly flexible partners. In addition, improvements in the Rosetta code is always welcome by the scientific community, and in particular for antibody design which is now of the highest relevance due to the coronavirus pandemic.

I have only one major comment that must be considered before publication and some minor comments to improve readability and reproducibility of the great work reported in this manuscript.

Major comment

Overall, the authors should compare docking results of antibody-antigen complexes to those retrieved by HADDOCK (https://doi.org/10.1016/j.str.2019.10.011) using the same benchmark of 16 complexes from the docking benchmark. It would be extremely useful to discuss those as well in the light of Rosetta/SnugDock performance because the aforementioned paper includes comparison of 4 different software. In addition, the authors should improve the illustration of data in all main figures and share their models, calculations and statistical analysis via the SBGrid database or another resource, in order for the work to be reproduced by others.

Response:

First and foremost, we thank Reviewer #1 for their comments. They have truly proven invaluable, highlight shortcomings in our descriptions and even catching a persistent typo (point 1-2).

Regarding the major concerns, we agree with the reviewer that a thorough comparison between antibody docking methods would be extremely useful. In fact, we have another publication that is currently under review where a detailed comparison between SnugDock, ClusPro, and ZDOCK is made on 67 complexes (Guest et al., revision under review for Structure; https://dx.doi.org/10.2139/ssrn.3564997). Readers should be able to make a fair comparison between this publication and the one cited above. We now include a proper discussion of two assessments in the introduction (lines 41-68):

Many software packages are available for protein–protein docking and several of them have modes specific for antibody–antigen docking; these include ClusPro [13, 14], FRODOCK [15], PatchDock [16], HADDOCK [17], and Rosetta SnugDock [18]. The first three methods are global, rigid-body approaches, adopting different docking algorithms. ClusPro and FRODOCK are fast-Fourier transformation (FFT) based. PatchDock decomposes proteins into geometric patches of hotspots and combines geometric hashing and pose clustering to identify interactions. On rigid targets, for which unbound structures are known, these methods tend to perform well. However, using homology models as input or docking flexible targets remains a challenge for these approaches. Methods such as HADDOCK and SnugDock were developed to address the challenge of flexible docking. HADDOCK is an information-driven flexible docking approach that combines a global rigid body search with ambiguous restraints, simulated annealing in torsion space, and minimization in Cartesian space. SnugDock is a local, flexible docking method that refines the CDR loops (including rebuilding the loops) and re-docks the VH–VL orientation in the context of the antibody–antigen interface. But to yield low-RMSD models, SnugDock requires an input orientation with the paratope close to the epitope, as it is not a global docking approach. Both HADDOCK and SnugDock were recently assessed with respect to other contemporary docking methods. HADDOCK was compared to ClusPro, LightDock, and ZDOCK on sixteen target complexes and generally out-performed the other methods [17]. HADDOCK achieved a 75% success rate (defined as having a model of acceptable quality or better in the top 10, according to the CAPRI quality definition [19]), whereas ClusPro acheived 67.8%. In another recent assessment, with 67 target complexes, ClusPro, ZDOCK, and SnugDock were compared. Therein, ClusPro achieved a 34% success rate (defined similarly) while SnugDock managed a 77.6% [20], but these rates are not directly comparable as SnugDock was benchmarked on local refinement, not a global search.

We additionally reference the two explicitly in lines 348-351.

A recent study comparing SnugDock, ZDOCK, and ClusPro is available [19]. Readers may also find useful another recent study comparing ClusPro, LightDock, ZDOCK, and HADDOCK [20].

In terms of data availability, all data and scripts have been made available on Zenodo (https://doi.org/10.5281/zenodo.4060853). All template models are available in versions of Rosetta from 2019, week 24 on. We have made this clear in the updated manuscript. We regret this initial oversight.

The models analyzed in this publication and the associated code are both available on Zenodo: doi.org/10.5281/zenodo.4060853.

Minor comments

1-Text

1.1-Page 5, line 132: section “New loop modeling and scientific benchmarks”. I understand the authors would like to publish those results elsewhere, but at least they should explain briefly what they did.

1-1 We have added the following sentences to provide more detail:

Briefly, the new loop modeling method expands on the default kinematic loop closure (KIC) modeling approach in Rosetta by allowing the sampling of backbone dihedral angles from homologous 3- and 9-residue fragments, in addition to the standard sampling from the Ramachandran plot, during loop modeling. After selecting appropriate fragments via the fragment picker, fragment KIC can be used during the loop modeling stages of RosettaAntibody and SnugDock. Additionally, we introduced a new form of testing alongside the standard unit and integration tests already present in the Rosetta code base. Scientific testing evaluates the performance of Rosetta on complete scientific tasks (e.g. running full antibody modeling simulations to test accuracy). We have made three antibody-related scientific tests: grafting, H3-loop modeling, and antigen docking. The tests are detailed in the subsections below and run automatically on our testing server https://benchmark.graylab.jhu.edu.

1.2-Page 6, line 156: The authors state that their previously published benchmark contains 48 targets, but in the previous publication 46 targets were included. I would recommend the authors to share along with Appendix S1 all pdb input files that were used for calculations as a zip file or deposit in SBGrid as recommended above.

1.2 – This is a typo and an incorrect citation; both have been corrected. The correct reference is https://www.jimmunol.org/content/jimmunol/early/2016/11/18/jimmunol.1601137.full.pdf and we actually have 47 antibodies in the set used to evaluated loop modeling on homology frameworks, omitting two antibodies 3MLR (atypically long CDR L3) and 1X9Q (single chain Fv). The PDB inputs are all released with Rosetta. The text has been updated correspondingly.

The grafting test runs the antibody executable for 47 targets (listed in the Appendix S1 and available in the antibody database that is distributed with Rosetta), a subset of the 49 targets originally described in Weitzner and Gray (2016) (we omit 3MLR due to its atypically long CDR-L3 loop and 1X9Q as it is an single chain antibody fragment), and it evaluates the RMSDs between the grafted models and the native crystal structures over all antibody structural regions (Table 1).

2-Figures.

2.1-Figure 1: Figure is good, but the old is shown of being of similar importance to the new FoldTree. Please show that the new one is the relevant one so that the reader knows directly by looking at the figure, that this is the one that was developed in this manuscript.

2-1 – Figure 1 and its caption have been updated to show the differences more clearly. To clarify to the reviewer, in the previous FoldTree one could only move the centers-of-mass of the constituent chains relative to one another (although there were ways around this, i.e. treating two chains as one in the code). With the new FoldTree, one can move individual chains or entire proteins relative to one another using any point in space as a center of mass, so while the FoldTree setup is more complex, docking itself requires no gimmicks.

2.2-Figure 2: Please include more details in the figure, explain what are the unique sequences for each plot in the legend; show the overlap of sequences in the old vs new databases with a venn diagram for each or another representation; show with additional figure panels example of interesting new data per category.

2.2 – Figure 2: We have replaced the bar plot with a set of Venn diagrams. We did not expand the figure with panels as there is no additional “interesting” data. This figure is meant to emphasize the growth in the number of templates, rather than highlight any particular new template. We now more clearly describe the figure in the caption.

The new database expands the number of unique sequences for which structural templates are available. For each structural region, the structures with unique sequences in the old versus new database is compared by Venn diagram. For all regions, there is substantial overlap. The new database always has more sequences than the old database. Some sequences are only found in the old database due to the consistent application of clear selection criteria in the automated database (whereas the previous database was manually curated).

2.3-Figure 3. Can you please make the figure legends more explanatory? Currently, they are called “Old Value” and “New value” instead of OCD (Å) or RMSD (Å). Results to my opinion, do not seem significant and authors cannot state that there is an improvement in grafted models as shown by these metrics. Authors should color-code the plots, showing with e.g. blue the grafted that were improved, black those with no change, and red those that worsened. In addition, statistics are missing, e.g. how many data points are per plot? Which exact models are included? The authors should share (or deposit in SBGrid as recommended above): (a) the old, the new grafted models and reference models plotted via the supplementary material as a zip file, (b) the script to calculate OCD so that plot results can be reproduced and (c) describe what is this exact RMSD that is calculated (main chain atoms? All chain? which exact regions?)

2.3 – Figure 3. We have expanded the figure to include axis labels for all subplots to be clearer (e.g. “BB RMSD (New) [Å]” instead of “new value”). Furthermore, we describe the metrics in the caption, “The OCD is a measure of the heavy-light chain orientation, previously described in Marze et al. The RMSD is calculated for backbone atoms for the region specified in the subfigure title and defined in Table 1”.

We do not want to claim that there are substantial improvements in performance, however it is important for us to report the average trend, so we have altered the language: “In the new database, 55% of CDR loops and 53.5% of FRs in the benchmark set have lower RMSD templates than in the old database.” We have color coded the figure, as requested. In the caption we specify, “Each subplot contains the results for the 47 targets listed in Appendix S1.” 

Finally, all models and calculated values are available in our Zenodo repository. RMSD and OCD are calculated by the antibody executable which is packaged and released with Rosetta.

2.4-Figure 4. Please change the colors of the figure because at the beginning I thought that plot B colors correspond to violin plot colors, which is not the case.

2.4 – Figure 4. The colors have been updated as requested.

3-Tables

3.1-Table 2. Is there a reason why on Table 2 there are ~300-400 templates removed from the old database? The authors should explain this in-text. What are these stringent quality criteria the authors mention in the Table legend? Only missing atoms or non-realistic C–N distances in critical regions? Please explain in-text or in a Table the old and the new criteria.

Considering the figure/table optimization and comparison to the other docking software, I believe this paper will be a seminal reference for future work regarding antibody design.

3.1 – Table 2. We clarify more explicitly why templates are removed between the two databases: 

Additionally, some sequences in the old database do not appear in the new database because it has more stringent quality criteria. Primarily, 307 structures do not meet the 3-angstrom resolution cutoff.

The quality criteria are now specified (before Figure 2 and Table 2):

We observed that not all the PDB IDs in the previous template database are retained in the new database, with the specific number varying by structural region. For the entire database, the difference amounts to 342 PDB IDs that are no longer included. This is due to a few key differences between the two databases. First, we ensure that entire antibody sequences are non-redundant in the new database, while no such check was present before. Second, we require that the resolution of structures in the new database is better than three angstroms (and of course this is only valid for X-ray structures). Again, no such cutoff was present before and NMR and EM structures were present. Of the 342 non-overlapping PDB IDs, this criterion accounts for the most, 307. Third, we have new quality criteria that were not previously implemented, requiring that (1) there are no C--N bond lengths of greater than 2 angstroms and CA--C--N and C--N--CA bond angels are within the ranges (89.5�, 144.5�) and (95�, 151�), respectively, (2) all templates can be loaded without error in Rosetta, (3) there are no 0 occupancy atoms in the PDB, and (4) no conserved framework residues, or residues used to identify the different antibody structural regions, are missing. For the overlapping portion of the two databases (identical PDBs), we compared the template structures and sequences to ensure that no drastic changes had occurred. The quality criteria exclude 19 of the 342, which leaves 16 to be excluded by the fact that these PDB records are now obsolete.

Reviewer #2: Reviewer Comments

The manuscript titled “ Robustification of RosettaAntibody and Rosetta SnugDock” describes developments made to the existing RosettaAntibody and Rosetta SnugDock softwares for antibody modelling and antigen-antibody docking, respectively. The advances made to each of these softwares are briefly summarized below.

For RosettaAntibody, the improvements include i. the automation of generating the template database by extracting PDB structures from another existing database- the Structural Antibody Database (SAbDab). The antibody structures extracted from the SAbDab, used for antibody modelling purposes, are numbered according to the Chothia scheme. ii. elimination of the requirement for a light chain during the grafting step of RosettaAntibody. ii. Minimizing user intervention by reducing the number of user-specified options during the homology modelling stage of RosettaAntibody and setting of default options (using the fragment-based kinematic loop closure (KIC) by default, to model C-term CDR-H3 insertions). iii. Options to exclude certain template PDB files or templates where the template and query sequences differ by the presence of proline residues in the CDR loops. iv. Incorporation of a glutamine-glutamine hydrogen bonding constraint during the modelling of the CDR H3 loop.

In case of SnugDock, the following changes made are i. Introduction of virtual residues and a modification to the existing kinematic representation (referred to as the “FoldTree”) to improve the flexibility of VH-VL and antigen-antibody docking motions. ii. automatic inclusion of the glutamine-glutamine hydrogen bonding constraint and kink constraint during the CDR-H3 modelling stage. iii. Docking of antigen to nanobodies. iv. Fragment-based loop modelling, as done for RosettaAntibody. v. Enabling usage of hydrogen exchange-mass spectrometry (HX-MS) data in the form of constraints to improve accuracy of the docking (detailed in another publication)

Apart from these modifications, the authors have also developed new scientific benchmarks for testing these methods, which is being described in another publication.

The authors tested the performance of the updated softwares using datasets published previously and compared the results with the same tests done on the earlier version of the softwares (including the earlier template database). The grafting step and loop modeling step of RosettaAntibody, and the docking step of SnugDock were tested.

The paper is well-written and balanced. Improvements as well as the limitations of the updated softwares have been detailed clearly. They have also described cases in which the updates did not have any significant improvement on the tests.

I have a few minor suggestions.

1. Does changing the default loop modelling step during CDR-H3 modelling to the kinematic loop closure method increase or decrease the time taken at the CDR-H3 modelling stage? The CDR-H3 modelling step has been tested on six antibody targets with a range of difficulties. Is it possible to run this step on slightly more number of antibody targets and make a comparison of the time (in addition to comparing the accuracy which has already been done) taken with the previous loop modelling and the kinematic loop closure method?This will give readers an idea of how long it would take to complete this specific step, as this is the limiting step during the modelling process.

2. Do the softwares always utilize all the processors it has been allocated, or does it sometimes use only a single processor despite multiple processors being allocated to it, if using a non-MPI version? What kind of system requirements will enable the non-MPI version of the softwares to make use of all the processors granted to it? A line or two about this could be included in the manuscript.

3. In Figure 5, the colors in the legend are not clear (at least in the PDF version), so maybe try to make the sizes a little larger.

Response:

We thank Reviewer #2 for their comments. 

1. To clarify, KIC has been the default loop modeling approach in RosettaAntibody as of 2016 (https://doi.org/10.4049/jimmunol.1601137). The time required to run KIC with and without fragments does not differ significantly. We have detailed the timing in another publication that is referenced: https://doi.org/10.1038/nprot.2016.180. 

2. Most Rosetta protocols are trivially parallelizable, so RosettaAntibody and SnugDock processes do not communicate with each other. This is documented online at: https://www.rosettacommons.org/docs/latest/rosetta_basics/MPI and https://www.rosettacommons.org/docs/latest/build_documentation/Build-Documentation. We now refer the reader to the “Getting Started” page which links to the above pages for building the software: 

“The Rosetta software can be accessed on https://www.rosettacommons.org and support for using Rosetta is available on https://www.rosettacommons.org/docs/latest/getting_started/Getting-Started.”

3. The figure has been updated with larger icons in the legend.

---

## [Decision Letter · Decision Letter 1]

11 Feb 2021

Robustification of RosettaAntibody and Rosetta SnugDock

PONE-D-20-14845R1

Dear Dr. Gray,

We’re pleased to inform you that your manuscript has been judged scientifically suitable for publication and will be formally accepted for publication once it meets all outstanding technical requirements.

Kind regards,

Horacio Bach

Academic Editor

PLOS ONE

Additional Editor Comments (optional):

Reviewers' comments:

Reviewer's Responses to Questions

**Comments to the Author**

1. If the authors have adequately addressed your comments raised in a previous round of review and you feel that this manuscript is now acceptable for publication, you may indicate that here to bypass the “Comments to the Author” section, enter your conflict of interest statement in the “Confidential to Editor” section, and submit your "Accept" recommendation.

Reviewer #1: All comments have been addressed

Reviewer #2: All comments have been addressed

Reviewer #3: All comments have been addressed

Reviewer #4: (No Response)

2. Is the manuscript technically sound, and do the data support the conclusions?

Reviewer #1: Yes

Reviewer #2: Yes

Reviewer #3: Yes

Reviewer #4: Yes

3. Has the statistical analysis been performed appropriately and rigorously? 

Reviewer #1: Yes

Reviewer #2: Yes

Reviewer #3: Yes

Reviewer #4: N/A

4. Have the authors made all data underlying the findings in their manuscript fully available?

Reviewer #1: Yes

Reviewer #2: (No Response)

Reviewer #3: Yes

Reviewer #4: Yes

5. Is the manuscript presented in an intelligible fashion and written in standard English?

Reviewer #1: Yes

Reviewer #2: Yes

Reviewer #3: Yes

Reviewer #4: Yes

6. Review Comments to the Author

Reviewer #1: The authors have addressed all my concerns and answered all my points. Congratulations for this solid manuscript!

Reviewer #2: All concerns raised and suggestions provided by the reviewers have now been appropriately addressed by the authors. This method hopes to be promising and widely applicable to the increasing antibody repertoire space.

Reviewer #3: Satisfied with detailed replies and changes to the manuscript. No additional comments.

Reviewer #4: (No Response)

7. PLOS authors have the option to publish the peer review history of their article (what does this mean?). If published, this will include your full peer review and any attached files.

Reviewer #1: No

Reviewer #2: No

Reviewer #3: No

Reviewer #4: No

---

## [Editor Report · Acceptance letter]

12 Mar 2021

PONE-D-20-14845R1 

Robustification of RosettaAntibody and Rosetta SnugDock 

Dear Dr. Gray:

I'm pleased to inform you that your manuscript has been deemed suitable for publication in PLOS ONE. Congratulations! Your manuscript is now with our production department. 

Kind regards, 

on behalf of

Dr. Horacio Bach 

Academic Editor

PLOS ONE